# Risk factors of early mortality among COVID-19 deceased patients in Addis Ababa COVID-19 care centers, Ethiopia

Taye Ashine Mezgebu[1]*, Migbar Mekonnen Sibhat[2], Melsew Tsegaw Getnet[3], Kassie Tiruneh Gebeyehu[2], Wuletaw Zewde Chane[3], Edmialem Mesfin Getahun[3], Asaminew Sane Habtamu[4], Hailu Beyene Asmare[5], Melke Mengistie Ambaw[6]

1 Schools of Nursing, College of Health Science and Medicine, Wachemo University, Hosanna, Ethiopia, 2 College of Health Science and Medicine, Dilla University, Dilla, Ethiopia, 3 Saint Paul's Hospital Millennium Medical College, Millennium COVID-19 Care Center, Addis Ababa, Ethiopia, 4 School of Nursing, College of Health Science and Medicine, Jimma University, Jimma, Ethiopia, 5 School of Nursing, College of Health Science and Medicine, Wolaita Sodo University, Sodo, Ethiopia, 6 Department of Nursing, Arba Minch College of Sciences, Arba Minch, Ethiopia

* tayemezgebu26@gmail.com

## Abstract

### Background

Severe acute respiratory syndrome coronavirus-2 is a global health care problem with high mortality. Despite early mortality seeming alarming, data regarding factors that lead to increased early mortality of COVID 19 patients is not well-documented yet. The objective of this study was to identify the risk factors of early mortality in patients with confirmed COVID-19 infections.

### Methodology

A case-control study design was employed. With this, a total of 261 COVID-19 deceased recordings were reviewed. The cases of the study were recordings of patients deceased within three days of intensive care unit admission whereas, the rest 187 were recordings of patients who died after three days of admission. Data were collected using an extraction checklist, entered into Epi data version 4.4.2.2, and analyzed by SPSS version 25. After the description, binary logistic regression was run to conduct bivariate and multivariable analyses. Finally, statistical significance was declared at p-value <0.05, and an adjusted odds ratio with a 95% confidence interval was used to report the strength of association.

### Result

The analysis was performed on 261 (87 cases and 174 controls) recordings. About 62.5% of the participants were aged above 65 years and two-thirds were males. The presence of cardiovascular disease (AOR = 4.79, with 95%CI: 1.73, 13.27) and bronchial-asthma (AOR = 6.57; 95% CI: 1.39, 31.13) were found to have a statistically significant association with early mortality. The existence of complications from COVID-19 (AOR = 0.22; 95% CI: 0.07,

**Data Availability Statement:** All relevant data are within the manuscript and its Supporting information files.

**Funding:** The authors received no specific funding for this work.

**Competing interests:** The authors have declared that no competing interests exist.

**Abbreviations:** AOR, Adjusted Odd Ratio; ARDS, Acute Respiratory Distress Syndrome; CI, Confidence Interval; COPD, Chronic Obstructive Pulmonary Disease; COVID-19, coronavirus disease-2019; DM, Diabetics Mellitus; ICU, Intensive Care Unit; IRB, Institutional Review Board; MRN, Medical Record Number; MSc, Master of Science; PEEP, Positive End Expiratory Pressure; SD, Standard Deviation; SPSS, Statistical Package for the Social Sciences; the USA, United States of America.

0.74) and previous history of COVID-19 infection (AOR = 0.17, 95% CI: 0.04, 0.69) were associated with decreased risk of early mortality.

## Conclusions

Having cardiovascular diseases and bronchial asthma was associated with an increased risk of early mortality. Conversely, the presence of intensive care unit complications and previous history of COVID-19 infection were associated with decreased risk of early mortality.

## Introduction

The emerged coronavirus disease-2019 (COVID-19), caused by the severe acute respiratory syndrome coronavirus-2 (SARS-CoV-2), has quickly spread worldwide with varied clinical characteristics [1]. Until 20th February 2022, the COVID-19 pandemic has affected about 422 million cases and about 5.8 million deaths globally, covering more than 220 countries worldwide [2]. However, Africa is forecasted to be the most defenseless continent where COVID-19 spreading will have a massive impact on the health system [3]. In the same period, Ethiopia is burdened with 469,047 cases and 7,477 deaths. The clinical presentation and the mortality rate are widely varied across the countries. In countries like Ethiopia, with a diverse population, the clinical profile of COVID-19 is different even among different regions.

Patients with coronavirus disease 2019 (COVID-19) are perceived to have an increased risk of mortality in the early days of ICU admission. Different epidemiological studies had evidenced that the mortality rate was higher among ICU admitted patients than non-ICU patients [4–6]. In the recent studies of COVID-19 patients from Wuhan, intensive care unit mortality rates ranged from 52% to 62% [7–10].

A study indicated that patients admitted early in the epidemic of COVID-19 had a higher mortality rate than patients admitted later [11]. Some studies among non-COVID-19 patients demonstrated that the risk factors of early mortality are different from determinants of late mortality [12, 13]. Shreds of evidence from plenty of studies showed that the risk factors regarding early mortality of COVID-19 are varied and may include male, older age, comorbidity, smoking, lymphocyte count, and D-dimer [13, 14]. It is also hypothesized that delay in ICU admission could be one of the factors for early mortality after ICU admission [15]. One-third of all patient deaths occur within three days of ICU admission [13].

Recognizing factors that can facilitate early mortality among patients admitted to the ICU settings is clinically significant. Identification of these factors aids in designing more effective ICU therapeutic strategies that aid to stabilize the patients' clinical condition, such as rescue therapy with advanced medical equipment and early human resources mobilization. Despite the clinical impact of early mortality is undisputable [11, 13, 16], studies assessing factors related to the early death of COVID-19 patients after ICU admission are scanty throughout the globe and in Ethiopia. Therefore, this study aims to identify risk factors for early mortality after ICU admission in patients with COVID-19.

## Methods and materials

A case-control study was carried out in Addis Ababa COVID-19 care centers, Ethiopia by reviewing charts of patients admitted from June 2020 to October 2021. There are three large COVID-19 care centers in Addis Ababa. Of those, Millennium COVID-19 care center is the

largest center, with a capability of 1000 beds, including 40 ICU beds in three separated ICU setups. The source population was all patients who deceased due to COVID-19 in the centers, whereas all COVID-19 confirmed patients who deceased in ICU from June 2020 to October 2021 were the study population. The cases of the study (early mortality) were recordings of patients deceased within three days of ICU admission whereas; the controls (late mortality) were recordings of patients who deceased after three days of admission. All recordings of COVID-19 patients who were deceased in the ICU from June 2020 to October 2021 with complete records were included. COVID-19 was classified in to mild, moderate, severe, and critical such that:

**Mild**: Patients present with fever, cough, fatigue, anorexia, myalgia and other non-specific symptoms without evidence of pneumonia.

**Moderate**: Patients present with fever, cough, dyspnea without severe pneumonia and hypoxia OR patients aged 60 years or above, OR with comorbidity.

**Severe/critical**: Patients having fever, cough, dyspnea with severe pneumonia or severe acute respiratory infection evidenced by RR>30breaths/minute, SpO2 <92%onroomair, OR signs of sepsis, organ failure, OR septic shock [17].

## Sample size determination

The sample size was determined using Epi-Info version 7.2 by considering diabetes mellitus (DM) as a determinant variable from a previous study [13] which yields the maximum sample size. The parameters used include a 95% confidence level, 80% study power, an odds ratio of 2.32, a 16.87% proportion of DM among controls, and a 1:2 case to control ratio. Finally, the total sample size was 264 (88 cases and 176 controls). Then, cases were selected intentionally by the researcher, and two respective controls next to the selected cases were selected randomly using the systematic random sampling technique.

## Data collection tools and procedures

Initially, the patient registries were evaluated to extract the total number of cases and to confine patients who died early and those who died late in the ICU. Then, the charts were arranged based on their medical record number (MRN) and the date of ICU admission. Afterward, the data were collected using a pretested and structured data extraction checklist which is designed from different literature [1, 13, 15, 18–23]. Then, the charts were arranged based on their medical record number (MRN) and the date of ICU admission. Afterward, the data were collected using a pretested and structured data extraction checklist which was newly designed with related variables. The checklist comprised socio-demographic characteristics, comorbidities, disease severity, baseline laboratory findings, and COVID-19 management-related variables. Six data collectors (MSc emergency and critical care nurses) and two supervisors were recruited and trained for one about the requirements of data collection.

## Analysis and interpretation of data

Data were cleaned, coded, and entered using Epi data version 4.4.2.2, and exported to SPSS version 25 for analysis. Cross-tabulation was done to determine the level of missing values and for data cleaning. Explorative analysis was conducted as well for continuous variables to determine the data distribution. Binary logistic regression was used to analyze the data. In the bivariate analysis, variables with a p-value of <0.2 were candidates for the final model for confounder adjustment. Multi-collinearity was checked using the variance inflation factor, and model fitness was checked using Hosmer and Lemeshow test (p = 0.361). In the multivariate

analysis, an adjusted odds ratio with 95% confidence intervals was computed to identify the presence and strength of associations. Statistical significance was stated at a p-value of <0.05.

## Ethical considerations

Ethical clearance was obtained from Saint Paul's Hospital Medical College institutional review board (IRB) (ref = PM23/799). The IRB waived the study to be conducted by chart review. Since patient consent was not applicable due to the retrospective nature of the study, permission for data collection was obtained from the clinical director. The responsible bodies at each intensive care unit and liaison office were informed about the purpose of the study. The study was conducted per the declaration of Helsinki. Confidentiality and anonymity were assured by data coding and aggregate reporting, where no individual patient identifiers were included.

## Results

This case-control study was conducted on 261 recordings of patients who died due to COVID-19 in the ICU (87 cases and 174 controls).

## Socio-demographic characteristics and comorbidities

The study finding showed that 62.5% of the study participants aged above 65 years. Of those, 28.8% died within the first 72 hours of ICU admission. The mean age of patients was 63.6years (SD = 15.7) among the cases and 68.6 years (SD = 13.7) among controls. On the other hand, about two-thirds (65.2%) of the patients were males, of which 32.4% passed away within three days of ICU admission (Table 1).

## Description of comorbidities and other clinical characteristics

According to the study results, the proportion of severe disease was higher (83.3%) among those who died late after ICU admission compared to patients who died early. On the contrary, a higher proportion of patients admitted with moderate disease (25.3%) died early after ICU admission than those who died after three days (12%). The study finding also revealed that 44.3% of the controls had a history of previous COVID-19 infection. Conversely, only 8% of patients with a history of COVID-19 infection were died within three days of ICU admission (early mortality). Regarding the preexisting medical illnesses, 183 (70.1%) of the patients had comorbidities, of which 83% had two or more comorbidities. The proportion of early mortality was much higher among patients with comorbid illnesses compared to those without comorbidities. The frequently documented chronic conditions were hypertension (35.6%), DM (35.2%), cardiovascular diseases (23.8%), and bronchial asthma (10.3%). Moreover, the proportion of patients who experienced COVID-19 complications was higher among the controls (77%) than in the early mortality (63%) (Table 2).

**Table 1. Socio-demographic characteristics between early mortality and late mortality among COVID-19 patients at Addis Ababa COVID-19 care centers, Ethiopia, 2022 (n = 261).**

| Variables | Category | Mortality | | Total (%) |
|---|---|---|---|---|
| | | Early (%) | Late (%) | |
| Age | <65 years | 40 (40.8) | 58 (59.2) | 98 (37.5) |
| | ≥65 years | 47 (28.8) | 116 (71.2) | 163 (62.5) |
| Sex | Male | 55 (32.4) | 115 (67.6) | 170 (65.2) |
| | Female | 32 (35.2) | 59 (64.8) | 91 (34.8) |

**Table 2. Comorbidities and other clinical related characteristics between early mortality and late mortality among COVID-19 patients at Addis Ababa COVID-19 care centers, Ethiopia, 2022 (n = 261).**

| Variables | Category | Early mortality (%) | Late mortality (%) | Total (%) |
|---|---|---|---|---|
| Smoking history | Yes | 65 (41.9) | 90 (58.1) | 155 (59.4) |
| | No | 22 (20.8) | 84 (79.2) | 106 (40.6) |
| Triage severity category | Mild | 2 (20.0) | 8 (80.0) | 10 (3.8) |
| | Moderate | 22 (51.2) | 21 (48.8) | 43 (16.5) |
| | Severe | 63 (30.3) | 145 (69.7) | 208 (79.7) |
| History of previous COVID-19 infection | Yes | 7 (8.3) | 77 (91.7) | 84 (32.2) |
| | No | 80 (45.2) | 97 (54.8) | 177 (67.8) |
| Number of comorbidities | None | 11 (14.1) | 67(85.9) | 78 (29.9) |
| | One | 10 (30.3) | 23 (69.7) | 33 (12.6) |
| | ≥ two | 66 (44.0) | 84 (56.0) | 150 (57.5) |
| Diabetics Mellitus | Yes | 42 (45.7) | 50 (54.3) | 92 (35.2) |
| | No | 45 (26.6) | 124 (73.4) | 169 (64.8) |
| Hypertension | Yes | 39 (41.9) | 54 (58.1) | 93 (35.6) |
| | No | 48 (28.6) | 120 (71.4) | 168 (64.4) |
| Cardiovascular disease | Yes | 38 (61.3) | 24 (38.7) | 62 (23.8) |
| | No | 49 (24.6) | 150 (75.4) | 199 (76.2) |
| Bronchial asthma | Yes | 14 (51.9) | 13 (48.1) | 27 (10.3) |
| | No | 73 (31.2) | 161 (68.8) | 234 (89.7) |
| COPD | Yes | 8 (100) | 0 | 8 (3.1) |
| | No | 79 (31.2) | 174 (68.8) | 253 (96.9) |
| Malignancy | Yes | 11 (37.9) | 18 (62.1) | 29 (11.1) |
| | No | 76 (32.8) | 156 (67.2) | 232 (88.9) |
| Renal disease | Yes | 5(50.0) | 5 (50.0) | 10 (3.8) |
| | No | 82 (32.7) | 169 (67.3) | 251 (96.2) |
| ICU admission delay | Yes | 39 (56.5) | 30 (43.5) | 69 (26.4) |
| | No | 48 (25.0) | 144 (75.0) | 192 (73.6) |
| Complication of COVID-19 | Yes | 55 (29.1) | 134 (70.9) | 189 (72.4) |
| | No | 32 (44.4) | 40 (56.2) | 72 (27.6) |

**Note**: COVID-19 complications include acute kidney injury, pulmonary embolism, shock, delirium, electrolyte disturbance, and pneumothorax

**Abbreviation**: ICU, Intensive Care Unit; COPD, chronic obstructive pulmonary disease; COVID-19-coronavirus 2019

### Description of baseline vital signs and laboratory findings

Concerning baseline vital signs, about 88.1%, 50.6%, and 73.2% of the patients were tachypneic, tachycardia, and hypoxic at admission. Among patients who died within 72 hours of ICU admission, 56.3%, 55.2%, and 31% had leukocytosis, anemia, and thrombocytopenia. The average white blood cell count at admission was $15.71\times10^3$c/μL (SD = 16.70) among the early mortality group and $14.82\times10^3$c/μL (SD = 11.30) among the controls. Furthermore, the mean serum albumin was 2.55g/dl (SD = 1.40) among patients who died within the first 72 hours of ICU admission and 2.6g/dl (SD = 0.92) among patients who died 3 days later after admission. Almost two-thirds of the early mortality had random blood sugar levels of >200mg/dl, whereas it was noted among just one-third of the controls (Table 3).

### COVID-19 management-related factors

Among the study participants, 62.1% of patients were intubated. Of those, 31% died within three days of ICU admission. The proportion of patients who received vasopressor therapy

**Table 3. Baseline vital sign parameters and laboratory finding between early mortality and late mortality among COVID-19 patients at Addis Ababa COVID care centers, Ethiopia, 2022 (n = 261).**

| Variables | Category | Cases (%) | Controls (%) | Total (%) |
|---|---|---|---|---|
| Respiratory rate (bpm) | <24 | 7 (22.6) | 24 (77.4) | 31 (11.9) |
| | ≥24 | 80 (34.8) | 150 (65.2) | 230 (88.1) |
| Systolic blood pressure (mmHg) | <90 | 6 (16.2) | 31 (83.8) | 37 (14.2) |
| | ≥90 | 81 (36.2) | 143 (63.8) | 224 (85.8) |
| Pulse rate beat per minute | <100 | 38 (29.5) | 91 (70.5) | 129 (49.4) |
| | ≥100 | 49 (37.4) | 83 (62.9) | 132 (50.6) |
| Oxygen saturation (%) | <90 | 68 (35.6) | 123 (64.4) | 191 (73.2) |
| | ≥90 | 19 (27.1) | 51 (72.9) | 70 (26.8) |
| Temperature (˚C) | <37.5 | 85 (35.9) | 152 (64.1) | 237 (90.8) |
| | ≥37.5 | 2 (8.3) | 22 (91.7) | 24 (9.2) |
| White blood cell count (cells/uL) | <11,000 | 38 (35.5) | 69 (64.5) | 107 (41.0) |
| | ≥ 11,000 | 49 (31.8) | 105 (68.2) | 154 (59.0) |
| Hemoglobin (milligram/dl) | <12 | 48 (49.0) | 50 (51.0) | 98 (37.5) |
| | ≥ 12 | 39 (23.9) | 124 (76.1) | 163 (62.5) |
| Platelet count (cells/uL) | <150,000 | 27 (42.9) | 36 (57.1) | 63 (24.1) |
| | ≥ 150,000 | 60 (30.3) | 138 (69.7) | 198 (75.9) |
| Creatinine, (milligram/dL) | ≤1.2 | 50 (25.6) | 145 (74.4) | 195 (74.7) |
| | > 1.2 | 37 (56.1) | 29 (43.9) | 66 (25.3) |
| Blood urea nitrogen (mg/dl) | <45 | 35 (26.9) | 95 (73.1) | 130 (49.8) |
| | ≥ 45 | 52 (39.7) | 79 (60.3) | 131 (50.2) |
| Alanine aminotransferase (U/L) | <40 | 45 (24.2) | 141 (75.8) | 186 (71.3) |
| | ≥ 40 | 42 (56.0) | 33 (44.0) | 75 (28.7) |
| Serum potassium (mmol/L) | <3.5 | 0 | 8 (100) | 8 (3.1) |
| | 3.5–5.0 | 60 (33.7) | 118 (66.3) | 178 (68.2) |
| | > 5 | 27 (36.0) | 48 (64.0) | 75 (28.7) |
| Serum sodium (mmol/L) | <135 | 22 (41.5) | 31 (58.5) | 53 (20.3) |
| | 135–146 | 65 (31.3) | 143 (68.8) | 208 (79.7) |
| Albumin (gram/dl) | <3.4 | 68 (33.2) | 137 (66.8) | 205 (78.5) |
| | ≥3.4 | 19 (33.9) | 37 (66.1) | 56 (21.5) |
| Random blood sugar (milligram /dl) | <200 | 31 (22.6%) | 106 (77.4) | 137 (525) |
| | ≥200 | 56 (45.2) | 68 (54.8) | 124 (47.5) |
| Chest radiological finding | Unilateral infiltration | 18 (19.8) | 73 (80.2) | 91 (34.9) |
| | Bilateral infiltration | 69 (40.6) | 101 (59.4) | 170 (65.1) |

**Abbreviation**: ICU, intensive care unit; bpm, breath per minute; ˚C, degree Celsius; dl, deciliter; mEq/L, mill equivalent per liter; mg/dl, mill gram per deciliter.

was higher among those who died late after ICU admission (59.8%) than those who died early (44.8%). Prone position was assumed by 40.2% of the cases and 74.1% of the controls. In addition, 95.1%, 95.4%, 85.4%, and 23.4% of the patients took systemic antibiotics, corticosteroid therapy, thromboprophylaxis, and antivirals respectively (Table 4).

## Risk factors of early mortality

After data description, bivariate and multivariable analyses were performed using the binary logistic regression model. Variables with a p-value of <0.25 in the bivariate analysis were candidates for confounder adjustment in the final model. After checking the model fitness, the

**Table 4. Treatment at ICU admission among early and late mortality of ICU admitted COVID-19 at Millennium COVID care center, Addis Ababa, Ethiopia 2022 (n = 261).**

| Variables | Category | Cases (%) | Controls (%) | Total (%) |
|---|---|---|---|---|
| Mode of ventilation | Non invasive | 37 (37.4) | 62 (62.6) | 99 (37.9) |
| | Intubated | 50 (30.9) | 112 (69.1) | 162 (62.1) |
| Broad-spectrum Antibiotics | Yes | 74 (29.8) | 174 (70.2) | 248 (95.1) |
| | No | 13 (100) | 0 (0%) | 13 (4.9) |
| Vasopressors, | Yes | 39 (27.3) | 104 (72.7) | 143 (54.8) |
| | No | 48 (40.7) | 70 (70.1) | 118 (45.2) |
| Systemic steroid | Yes | 81 (32.5) | 168 (67.5) | 249 (95.4) |
| | No | 6 (50.0) | 6 (50.0) | 12 (4.6) |
| Thromboprophylaxis | Yes | 69 (30.9) | 154 (69.1) | 223 (85.4) |
| | No | 18 (47.4) | 20 (52.6) | 38 (14.6) |
| Antihypertensive | Yes | 19 (32.2) | 40 (67.8) | 59 (22.6) |
| | No | 68 (33.7) | 134 (66.3) | 202 (77.4) |
| Antiviral | Yes | 11 (18.0) | 50 (82.0) | 61 (23.4) |
| | No | 76 (38.0) | 124 (62.0) | 200 (76.6) |
| Prone position | Yes | 35 (21.3) | 129 (80.0) | 164 (62.8) |
| | No | 52 (53.6) | 45 (46.4) | 97 (37.2) |
| Blood transfusion | Yes | 0 | 29 (100) | 29 (11.1) |
| | No | 87 (37.5) | 145 (62.5) | 232 (88.8) |
| Hemodialysis | Yes | 11 (20.0) | 44 (80.0) | 55 (21.1) |
| | No | 76 (36.9) | 130 (63.1) | 206 (78.9) |

multivariable analysis was run. In the multivariate analysis, COVID19 complications, cardiovascular disease (CVD), bronchial asthma, and previous history of COVID-19 infection showed statistically significant association with early mortality of COVID-19 patients at a 95% confidence level.

Patients having CVD were 4.79 times (AOR = 4.79; 95%CI: 1.73, 13.27) more likely to die within the first 72 hours of ICU admission compared to those who had no CVD. Likewise, patients with bronchial asthma had 6.57 times increased chance of early mortality compared to their counterparts (AOR = 6.57; 95% CI: 1.39, 31.13). On the contrary, the odds of early mortality among patients who had a history of previous COVID-19 infection were 83% decreased compared to their counterparts (AOR = 0.17; 95% CI: 0.04, 0.69). The other variable that showed significant association with early mortality was the presence of COVID-19 complications. Patients who experienced COVID-19 complications (such as acute kidney injury, pulmonary embolism, shock, delirium, electrolyte disturbance, and pneumothorax) were 78% times less likely to die within the first three days than after then (Table 5).

## Discussion

Existing evidence showed that several COVID-19 patients died within the first three days of admission [16, 24]. Identifying the factors for this increased death in the early stages of ICU admission would enable us to set evidence-based interventions and reduce this problem. Perhaps, to the best of researchers' knowledge, no such study has been conducted in Ethiopia. Thus, this case-control study assessed the risk factors of early mortality among COVID-19 patients at the intensive care unit of Addis Ababa COVID-19 care centers. After adjusting for potential confounders, four variables were found to be independent risk factors of early mortality among COVID-19 ICU patients.

**Table 5. Bivariate and multivariate analysis of risk factors associated with early mortality after ICU admission among COVID-19 patients at Addis Ababa COVID 19 care centers, Ethiopia, 2022 (n = 261).**

| Risk factors | Category | Mortality | | COR (95%CI) | AOR (95%CI) | Sig. |
|---|---|---|---|---|---|---|
| | | Late | Early | | | |
| Age | <65 years | 58 | 40 | 1 | 1 | 1 |
| | ≥65 years | 116 | 47 | 0.59 (0.35, 0.99) | 0.51 (0.19, 0.1.32) | 0.165 |
| Triage score | Not severe | 24 | 29 | 1 | 1 | 1 |
| | Severe | 63 | 145 | 0.53 (0.28–0.97) | 0.48 (0.18, 1.27) | 0.138 |
| Complication of COVID-19 | Yes | 134 | 55 | 1.95 (1.11, 3.42) | 0.22 (0.08, 0.59) | **0.003**[*] |
| | No | 40 | 32 | 1 | 1 | 1 |
| Prone position | Yes | 129 | 35 | 0.24 (0.14, 0.41) | 0.56 (0.15, 2.08) | 0.384 |
| | No | 45 | 52 | 1 | 1 | |
| History of previous COVID-19 infection | Yes | 77 | 7 | 9.07 (0.3.96, 20.77) | 0.17 (0.04, 0.69) | **0.013**[*] |
| | No | 97 | 80 | 1 | 1 | 1 |
| Diabetes Mellitus | Yes | 29 | 81 | 2.32 (01.36, 3.95) | 0.96 (0.34, 2.68) | 0.934 |
| | No | 28 | 32 | 1 | 1 | 1 |
| Hypertension | Yes | 25 | 34 | 1.82 (.94,3.51) | 1.65 (0.53, 5.13) | 0.388 |
| | No | 32 | 79 | 1 | 1 | |
| Cardiovascular disease | Yes | 24 | 14 | 4.85 (2.65, 8.87) | 4.79 (1.73, 13.27) | **0.003**[*] |
| | No | 33 | 99 | 1 | 1 | |
| Bronchial asthma, | Yes | 13 | 14 | 2.38 (1.06, 7.53) | 6.569 (1.39, 31.13) | **0.018**[*] |
| | No | 161 | 73 | 1 | 1 | 1 |
| ICU admission delay | Yes | 30 | 39 | 0.26 (0.14, 0.46) | 1.41 (0.53, 3.71) | 0.491 |
| | No | 144 | 48 | 1 | 1 | |
| Vasopressor | Yes | 104 | 39 | 0.55 (0.33, 0.92) | 0.85 (0.33, 2.18) | 0.728 |
| | No | 70 | 48 | 1 | 1 | 1 |
| Pulse rate (beat per minute) | <100 | 91 | 38 | 1 | 1 | 1 |
| | ≥100 | 83 | 49 | 1.41 (0.84, 2.37) | 2.89 (0.92, 9.12) | 0.070 |
| Peripheral oxygen saturation (%) | <90 | 123 | 68 | 1.48 (0.811, 2.716) | 0.52 (0.12, 2.19) | 0.372 |
| | ≥90 | 51 | 19 | 1 | 1 | 1 |
| Creatinine (mg/dl) | <1.2 | 145 | 50 | 1 | 1 | 1 |
| | ≥ 1.2 | 29 | 37 | 3.70 (2.07, 6.63) | 1.91 (0.48, 7.53) | 0.358 |
| Platelet count (cells/μl) | <150,000 | 36 | 27 | 1.73 (0.95, 3.09) | 1.52 (0.56, 4.10) | 0.414 |
| | ≥ 150,000 | 138 | 60 | 1 | 1 | 1 |
| Alanine amino-transferase (U/L) | <40 | 141 | 45 | 1 | 1 | 1 |
| | ≥ 40 | 33 | 42 | 2.29 (1.23, 4.28) | 1.30 (0.44, 3.86) | 0.632 |
| Hemoglobin (g/dl) | <12 | 50 | 48 | 3.05 (1.79, 5.21) | 1.12 (0.38, 3.29) | 0.840 |
| | ≥ 12 | 124 | 39 | 1 | 1 | 1 |
| Serum sodium (mmol/L) | >135 | 31 | 22 | 1.56 (0.84, 2.90) | 1.72 (0.51, 5.80) | 0.385 |
| | 135–145 | 143 | 65 | 1 | 1 | 1 |
| Random blood sugar (mg/dl) | <200 | 106 | 31 | 1 | 1 | 1 |
| | ≥200 | 68 | 56 | 0.37 (0.19, 0.72) | 2.09 (0.59, 7.39) | 0.253 |

Note

[*] significantly associated with early mortality

**Note**: COVID-19 complications include acute kidney injury, pulmonary embolism, shock, delirium, electrolyte disturbance, and pneumothorax

**Abbreviation**: COR, crude odds ratio; AOR, adjusted odds ratio; CI, confidence interval; COVID-19, coronavirus disease 2019; ICU, intensive care unit

Patients having CVD were 4.79 times (AOR = 4.79; 95%CI: 1.73, 13.27) more like to die within the first 72 hours of ICU admission compared to those who had no CVD. This finding is consistent with the study conducted in Pakistan [18] and Wuhan, China [8, 25]. However, these studies investigated the factors associated with mortality of COVID-19 patients as a whole and were not specific to early mortality. The possible justification could be the effect of COVID-19 on the cardiovascular system. Severe COVID-19 disease is associated with hemodynamic instabilities, severe ARDS (requiring high PEEP levels of ventilation), decreased cardiac output, hypervolemia, sepsis, and those who received nephrotoxic drugs [26]. These effects may, in turn, worsen the preexisting cardiovascular disease leading to more severe illness, cardiac arrest, and higher mortality [24].

Likewise, patients with bronchial asthma had 6.57 times increased chance of early mortality compared to their counterparts (AOR = 6.57; 95% CI: 1.39, 31.13). This finding was conformable with previous studies in the United Kingdom [27] and Korea [28] that reported a higher risk of COVID-19 severity and mortality among asthmatic patients. The reason for this might be that COVID-19 infection trigger acute exacerbation of bronchial asthma that leads to oxygen hunger. Hence, hypoxia can result in respiratory failure ending up in premature death. However, the findings of studies conducted in France [29] and New York City [30] contradict the current study result. This discrepancy might be due to the difference in the study subjects. Unlike in our study, previous studies reported factors for overall COVID-19 mortality (merging both late and early mortality). Further studies are therefore required to completely elucidate the role of asthma in early mortality.

On the contrary, the odds of early mortality among patients who had a history of previous COVID-19 infection were 83% decreased compared to their counterparts (AOR = 0.17; 95% CI: 0.04, 0.69). This finding is supported by studies done in the USA [31], Australia [32], and the UK [33] that showed a low reinfection rate, morbidity, and mortality among those who had a history of previous COVID-19 infection. This might be due to the development of antibodies and strengthened immune systems after being infected by COVID-19, which declines the probability of severe disease [34, 35]. Even though we could not find former reports regarding the association between early mortality and previous history of COVID-19 infection, this finding magnifies the impact of COVID-19 vaccination to boost immunity and patient outcome.

The other variable that showed significant association with early mortality was the presence of COVID-19 complications. Patients who experienced COVID-19 complications had a 78% reduced mortality risk within the first 72 hours than after three days of ICU admission. The rationale could be that most fatal ICU complications such as septic shock, acute kidney injury, hospital-acquired infections, and ventilator-associated pneumonia occur late in the ICU (after 48–72 hours of admission) after intubation [36]. Furthermore, even patients who developed ICU complications in the early days may die after three days and recorded as late mortality. Hence, this high record of complication-associated deaths in the late mortality group might conceal the effect of complications on the early mortality group despite its impact on poor patient outcomes, in general, being undisputable. To strengthen this rationale, deaths linked to COVID-19 complications were more prevalent among the late mortality group in this study.

While using and interpreting the findings of this study, the following issues should be taken into consideration. The main limitation of this study was the lack of consideration of the duration from symptom onset up to hospital admission. Irrespective of the possible factors for early mortality, patients who had stayed longer to seek medical attention might not have a similar length of hospital stay and in-hospital outcome compared to those who did not delay. Secondly, as the samples were selected based on the outcome status, it could be quite difficult to evaluate the effect of some variables (e.g. ICU complications) that might affect the survival and

timing of mortality. Besides, the general limitation of retrospective studies could be another point to be reminded of while applying the study findings.

## Conclusion

Having cardiovascular diseases and bronchial asthma was associated with an increased risk of early mortality. Conversely, the presence of ICU complications and previous history of COVID-19 infection were associated with reduced risk of mortality in the first three days of ICU admission. Hence, it is essential to put specific strategies to manage and control the identified factors to minimize the high burden of early mortality among COVID-19 patients.

## Supporting information

**S1 Data.**
(SAV)

## Acknowledgments

The center administrators, staff, data collectors, and supervisors were appreciated for providing the necessary preliminary information. The authors would also like to thank Saint Paul's Hospital Millennium Medical College to pursue the chance to conduct this study.

## Author Contributions

**Conceptualization:** Taye Ashine Mezgebu, Melsew Tsegaw Getnet, Wuletaw Zewde Chane, Asaminew Sane Habtamu, Hailu Beyene Asmare, Melke Mengistie Ambaw.

**Data curation:** Taye Ashine Mezgebu, Migbar Mekonnen Sibhat, Kassie Tiruneh Gebeyehu, Wuletaw Zewde Chane, Edmialem Mesfin Getahun, Asaminew Sane Habtamu, Hailu Beyene Asmare, Melke Mengistie Ambaw.

**Formal analysis:** Taye Ashine Mezgebu, Migbar Mekonnen Sibhat, Melsew Tsegaw Getnet, Kassie Tiruneh Gebeyehu, Edmialem Mesfin Getahun, Asaminew Sane Habtamu, Hailu Beyene Asmare, Melke Mengistie Ambaw.

**Funding acquisition:** Melsew Tsegaw Getnet, Wuletaw Zewde Chane, Edmialem Mesfin Getahun.

**Investigation:** Taye Ashine Mezgebu, Migbar Mekonnen Sibhat, Melsew Tsegaw Getnet, Kassie Tiruneh Gebeyehu, Wuletaw Zewde Chane, Edmialem Mesfin Getahun.

**Methodology:** Taye Ashine Mezgebu, Migbar Mekonnen Sibhat, Melsew Tsegaw Getnet, Wuletaw Zewde Chane, Asaminew Sane Habtamu, Hailu Beyene Asmare, Melke Mengistie Ambaw.

**Project administration:** Taye Ashine Mezgebu, Melsew Tsegaw Getnet, Wuletaw Zewde Chane, Edmialem Mesfin Getahun.

**Resources:** Melsew Tsegaw Getnet, Wuletaw Zewde Chane, Edmialem Mesfin Getahun, Hailu Beyene Asmare.

**Software:** Taye Ashine Mezgebu, Migbar Mekonnen Sibhat, Melsew Tsegaw Getnet, Kassie Tiruneh Gebeyehu, Asaminew Sane Habtamu, Melke Mengistie Ambaw.

**Supervision:** Taye Ashine Mezgebu, Migbar Mekonnen Sibhat, Melsew Tsegaw Getnet, Kassie Tiruneh Gebeyehu, Wuletaw Zewde Chane, Edmialem Mesfin Getahun, Asaminew Sane Habtamu, Hailu Beyene Asmare, Melke Mengistie Ambaw.

**Validation:** Taye Ashine Mezgebu, Migbar Mekonnen Sibhat, Melsew Tsegaw Getnet, Kassie Tiruneh Gebeyehu, Wuletaw Zewde Chane, Edmialem Mesfin Getahun, Asaminew Sane Habtamu, Hailu Beyene Asmare, Melke Mengistie Ambaw.

**Visualization:** Taye Ashine Mezgebu, Migbar Mekonnen Sibhat, Melsew Tsegaw Getnet, Kassie Tiruneh Gebeyehu, Wuletaw Zewde Chane, Edmialem Mesfin Getahun, Asaminew Sane Habtamu, Hailu Beyene Asmare, Melke Mengistie Ambaw.

**Writing – original draft:** Taye Ashine Mezgebu, Migbar Mekonnen Sibhat, Kassie Tiruneh Gebeyehu, Hailu Beyene Asmare.

**Writing – review & editing:** Migbar Mekonnen Sibhat, Melsew Tsegaw Getnet, Kassie Tiruneh Gebeyehu, Wuletaw Zewde Chane, Edmialem Mesfin Getahun, Asaminew Sane Habtamu, Hailu Beyene Asmare, Melke Mengistie Ambaw.

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
