## [Decision Letter · Decision Letter 0]

5 Jul 2022

PONE-D-22-10705Risk Factors of early mortality among COVID-19 deceased patients in Addis Ababa COVID-19 Care Centers, EthiopiaPLOS ONE

Dear Dr. Mezgebu,

Thank you for submitting your manuscript to PLOS ONE. After careful consideration, we feel that it has merit but does not fully meet PLOS ONE’s publication criteria as it currently stands. Therefore, we invite you to submit a revised version of the manuscript that addresses the points raised during the review process.

We look forward to receiving your revised manuscript.

Kind regards,

Zivanai Cuthbert Chapanduka, MBChB (M.D)

Academic Editor

PLOS ONE

Journal Requirements:

Additional Editor Comments:

Thank you for the manuscript.

Please read the reviewers comments and requirements carefully and respond to the best of your ability

Reviewers' comments:

Reviewer's Responses to Questions

**Comments to the Author**

1. Is the manuscript technically sound, and do the data support the conclusions?

Reviewer #1: Yes

Reviewer #2: Yes

2. Has the statistical analysis been performed appropriately and rigorously? 

Reviewer #1: Yes

Reviewer #2: Yes

3. Have the authors made all data underlying the findings in their manuscript fully available?

Reviewer #1: Yes

Reviewer #2: Yes

4. Is the manuscript presented in an intelligible fashion and written in standard English?

Reviewer #1: Yes

Reviewer #2: Yes

5. Review Comments to the Author

Reviewer #1: Thank you for the opportunity to review this paper.We have a few comments/questions for the authors.

1 Were participants and/or researchers blind?

2. On what type of significance is the focus in the presentation of results: statistical or clinical?

3. Line 90:Delete despite, start with International evidence suggests that ....and find some references

4. When was the diagnosis made, that is important before ICU admission

5. Please explain, what you mean by intentionally, as you also mention randomly selecting the patients.

6. Line 161: delete have

7. Line 158: what parameters were used as mild, moderate and severe disease?

8. Line 163: This sentence is a bit confusing, please rephrase, I assume you are trying to say whereas only 8% of patients with early mortality ......

9. Line 169: rephrase to than in the cases with early mortality

10: Table 2: What parameters were looked with regards to complications of COVID-19.

11. Line 181: Reword: cases to with early mortality

12. Line 184: Reword cases to: early mortality case208: typing error most likely: please change like to likely

Line 216: What parameters did you look at as "COVID complications"? Please name them however leave it under one group as mentioned in the table.

13: Sentence that starts in line 227: Is this statement completely true? Perhaps no such study has been conducted in Ethiopia would be more appropriate.

14. Last sentence beginning in line 252: Rephrase, to additional studies are therefore required to completely elucidate the role of asthma in early mortality or leave it out completely as you have discussed why other findings are different from yours.

15: Line 270: delete counted and rather say were recorded or were grouped under the late mortality group based on the criteria of early vs late mortality.

Reviewer #2: Abstract:

Is this study a Case-Control? All patients have the disease, therefore there is no ‘Control’ group without the disease. Consider re-classifying the study type. This study is a retrospective review of patient files, the authors need to be very clear about this.

What is the ‘response rate’ referring to? This normally applies to survey type studies using questionnaires. Consider the use of this term

Introduction

Line 74 is ambiguous, consider revising the last phrase: ‘‘…mortality in the early days of ICU admission before the causes of death are recognized’’

Ethics Approval number should be included in the manuscript.

Use of the term ‘’response rate’’ also requires revision. this normally applies to qualitative/survey-type studies and does not seem to apply to this study type.

Methods: study design needs revision and clarity.

Discussion

Line 229: the word 'lately' should be later

Line 281: samples recruited does not sound right, you cannot recruit samples.

Conclusion: Does an association with late mortality necessarily imply protection from early mortality? the researchers also need to be able to clarify how severe disease with complications could be physiologically protective; what physiological or clinical management factors could explain this finding?

Line 289: avoid using the word 'halt': chronic diseases cannot easily be halted, their course tends to be slowly progressive.

6. PLOS authors have the option to publish the peer review history of their article (what does this mean?). If published, this will include your full peer review and any attached files.

Reviewer #1: **Yes: **Dr Ravnit Grewal MBChB, FC Path (Haem) , M Med (Stellenbosch University), Principal Pathologist/Senior Lecturer, Department of Haematology and Cell Biology, Faculty of Health Sciences, National Health Laboratory Services/ University of the Free State Bloemfontein, South Africa

Reviewer #2: No

---

## [Author Response · Author response to Decision Letter 0]

31 Jul 2022

Point-by-point responses were provided and attached as a separate file in the submission system.

---

## [Decision Letter · Decision Letter 1]

12 Sep 2022

Risk Factors of early mortality among COVID-19 deceased patients in Addis Ababa COVID-19 Care Centers, Ethiopia

PONE-D-22-10705R1

Dear Dr. Mezgebu,

We’re pleased to inform you that your manuscript has been judged scientifically suitable for publication and will be formally accepted for publication once it meets all outstanding technical requirements.

Kind regards,

Zivanai Cuthbert Chapanduka, MBChB (M.D)

Academic Editor

PLOS ONE

Additional Editor Comments (optional):

Reviewers' comments:

Reviewer's Responses to Questions

**Comments to the Author**

1. If the authors have adequately addressed your comments raised in a previous round of review and you feel that this manuscript is now acceptable for publication, you may indicate that here to bypass the “Comments to the Author” section, enter your conflict of interest statement in the “Confidential to Editor” section, and submit your "Accept" recommendation.

Reviewer #1: (No Response)

Reviewer #2: All comments have been addressed

2. Is the manuscript technically sound, and do the data support the conclusions?

Reviewer #1: (No Response)

Reviewer #2: Yes

3. Has the statistical analysis been performed appropriately and rigorously? 

Reviewer #1: (No Response)

Reviewer #2: Yes

4. Have the authors made all data underlying the findings in their manuscript fully available?

Reviewer #1: (No Response)

Reviewer #2: No

5. Is the manuscript presented in an intelligible fashion and written in standard English?

Reviewer #1: (No Response)

Reviewer #2: Yes

6. Review Comments to the Author

Reviewer #1: Thank you for the opportunity to edit the manuscript. It is indeed a topical subject. I wish you all the best in your future research and would hope that you have follow up studies on this project as well.

Reviewer #2: Thank you for the revisions.

The manuscript reads well. The authors should also double-check the final version of the manuscript since the Journal does not edit manuscripts.

7. PLOS authors have the option to publish the peer review history of their article (what does this mean?). If published, this will include your full peer review and any attached files.

Reviewer #1: **Yes: **Dr Ravnit Grewal

Reviewer #2: No

---

## [Editor Report · Acceptance letter]

19 Sep 2022

PONE-D-22-10705R1 

Risk Factors of early mortality among COVID-19 deceased patients in Addis Ababa COVID-19 Care Centers, Ethiopia 

Dear Dr. Mezgebu:

I'm pleased to inform you that your manuscript has been deemed suitable for publication in PLOS ONE. Congratulations! Your manuscript is now with our production department. 

Kind regards, 

on behalf of

Dr. Zivanai Cuthbert Chapanduka 

Academic Editor

PLOS ONE